# Dynamic Extrusion Control in Spot Deposition Modeling for Porous 3D Clay Structures

Vesela Tabakova *, Christina Klug  and Thomas H. Schmitz

Department of Visual Arts, Faculty of Architecture, RWTH Aachen University, Schinkelstr. 1,
52062 Aachen, Germany; klug@kg.rwth-aachen.de (C.K.); schmitz@kg.rwth-aachen.de (T.H.S.)
* Correspondence: tabakova@kg.rwth-aachen.de

**Abstract:** The dynamic state of the viscous clay in Liquid Deposition Modeling (LDM) often leads to discrepancies between the digital model and the resulting physical object. This emergent behavior can be harnessed to produce complex physical structures that would not be possible with other methods. This study takes advantage of the viscous state and tensile strength of the extruded clay strand to explore the impact of dynamic extrusion and deformations through travel paths in LDM to manufacture complex porous physical structures. The effects of these parameters are discussed in two case studies: (1) regular and semi-random Spot Deposition surfaces with either open or thickened regions, and (2) porous 3D lattice structures created through the controlled bending of vertical extrusions. The achieved higher geometrical complexity of objects through the algorithmically programmed alternations in the sequence and rate of material deposition allows for a wide range of buildup approaches that expand the production spectrum of sustainable small- and large-scale elements.

**Keywords:** LDM; Spot Deposition; emergent material behavior; 3D clay printing



## 1. Introduction

Most conventional 3D printers today have an extruder motor that allows for digital control of the material extrusion, enabling varying amounts of material deposition based on preset rules determined by the desired geometry [1]. This aspect of the printing strategy has already been explored in Fused Deposition Modeling (FDM) and lies behind some of the recent advances in slicing software [2,3]. However, these slicing algorithms developed for FDM are only vaguely applicable in Liquid Deposition Modeling (LDM) due to differences between the properties of thermoplastic filament and viscous buildup materials, such as clay paste [4,5]. The fluid behavior of clay paste produces discrepancies during deposition and shrinkage when drying [6,7]. This underscores the importance of a meticulous approach in the digital design process, where these emergent characteristics are used as design tools [8]. One of the already studied effects is the deformation of the extruded clay coil under self-weight to produce surface effects or surface porosity [9,10]. The consistency of the clay paste for LDM has also been used to selectively re-extrude material in already deposited layers, leading to better adhesion between them [11]. However, these studies are enabled only through digitally generated extrusion paths. Additive Manufacturing (AM) processes through actuator material deposition also consist of material flow parameters and travel paths where no material is deposited, which adds control over the rate and geometry of material deposition. This, in turn, enables the creation of geometric complexity in the physical object without the need for complex digital volume models.

The effects of these parameters in clay paste deposition have only been marginally explored until now. Simple ceramic lattice structures have been produced on the micro-scale using Direct Ink Writing (DIW) [12,13]. This has been achieved by creating layers with alternating path directions, where the extruded strand spans freely between two material

strands from a lower layer, creating a tetragonal geometry. This produces lattice structures consisting of rectangular open pores with maximum dimensions of $2.4 \times 2.4 \times 1.0$ mm. Using this additive manufacturing strategy, the height of the pore size is limited to the layer height, and the width and length are limited to the maximum span that could be achieved before the sagging of the extruded material strand. Further, the scale of the produced porous objects using DIW is a magnitude smaller than the scale of the objects produced using LDM, which makes structural properties hard to compare. One example of material deformation and varying volumes of extruded material per traveled path (dynamic extrusion) being utilized is the study Spatial Print by S. AlOthman [14], where the creation of lattice structures using a robotic arm with six degrees of freedom (DoF) was explored. A 3D lattice path was created through vertical spatial loops. These loops were enabled partly through the folding of the clay strand, but mostly through a higher accumulation of material in the vertical deposition path of the loop. The tool path and material properties enabled the creation of a structure with 3D open pores. The porosity of these lattice structures was evaluated by calculating the porosity area of the main vertical faces, where the best structurally performing loop pattern had a face porosity of 25% and pore size of 30 mm width and 15 mm height. However, due to sagging under self-weight and settling due to the weight of subsequent clay layers, the physical model deviated from the digital extrusion path. Further research deployed a closed-loop system consisting of a robotic arm that deposits material, heats up, scans and adjusts the digital model to mitigate the deviations caused by the plasticity of the material [15]. The project succeeded in creating a three-dimensional lattice structure using viscous clay. However, the complexity of the deployed system is non-trivial and computationally intensive.

Spot Deposition (SD) in AM relates to the act of material deposition, where the actuator first travels to the desired position and then deposits a discrete amount of material. This differentiates from the conventional FDM, where the material is continuously extruded while the actuator moves. The SD strategy is applied in metal AM, where sections of metal wire filament are molten into droplets that can be layered (spot welding) to manufacture metal lattice structures [16,17]. There are also precedent projects on the architectural scale from Gramazio Kohler Research, where uniformly pre-cut loam cylinders with a diameter of 9 cm and a height of 15 cm are robotically deposited into a predetermined position through ballistic projection [18] or pressed at a discrete point to create a large scale three-dimensional wall structure [19,20]. In both cases, the additional force provided by the impact, or the mechanical pressure, allows for the creation of pre-compressed structures that are less prone to settling. A disadvantage of this method is that, due to the added pressure, no geometries with overhang can be produced.

In this paper, the plastic and viscous behaviors of clay paste in combination with digitally generated non-horizontal tool paths and dynamic extrusion are utilized as a novel AM strategy for clay LDM to allow for precise modeling of ceramic structures with different porosities. Complex porous ceramic structures can find application in ventilation, acoustics, scaffolding for organic or inorganic matter on different scales and for a wide range of use cases in technical engineering and construction [21].

## 2. Materials and Methods

### 2.1. Clay Paste

This study utilized a stoneware clay body comprising 25% chamotte with grain sizes ranging from 0 to 0.5 mm (Table 1).

The clay paste was prepared manually by drying and rehydrating the clay body to achieve an optimal consistency for extrusion. This process led to fluctuating plasticity during the experiments. The water content of the paste was measured by taking samples between tests of equal amount, i.e., the same amount of motor rotations and with same tank air pressure (see Section 2.2). The wet sample weight was compared to the weight after drying at 100 °C for 24 h. The average water content of the used clay paste was 21.63 wt.% with a fluctuation of $\pm 1.64$ wt.% (Figure 1).

**Table 1.** Technical data * for the "Ateliermasse Weiß 2505" stoneware body.

| Chemical Analysis | SiO$_2$ | TiO$_2$ | Al$_2$O$_3$ | Fe$_2$O$_3$ | CaO | MgO | K$_2$O | Na$_2$O |
|---|---|---|---|---|---|---|---|---|
| | 75.0% | 1.4% | 19.5% | 0.80% | 0.20% | 0.30% | 2.30% | 0.20% |
| Moisture | 16.8wt.% | | | | | | | |

* Data retrieved from SIBELCO [22]. Data for the stoneware body refer to unfired condition.

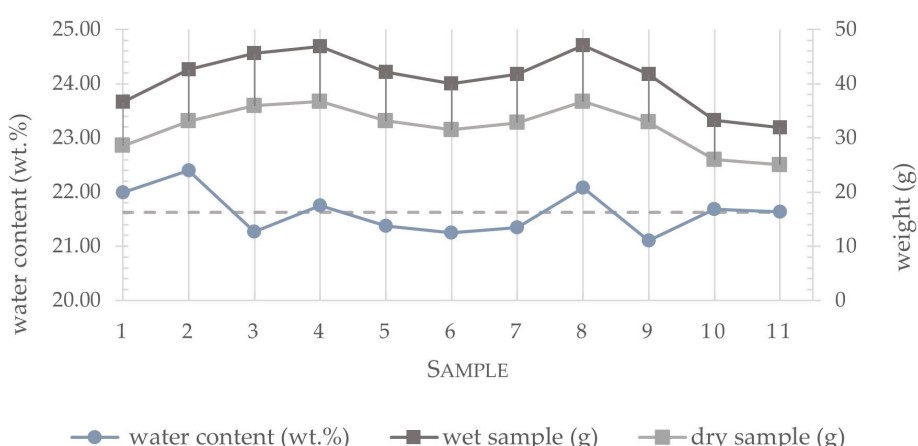

**Figure 1.** Water content (wt.%) of the clay paste compared to the wet and dry sample weights.

*2.2. Liquid Deposition Modeling System*

AM is a process in which successive material deposition creates a physical object from a digital file. Liquid Deposition Modeling (LDM—first coined by World's Advanced Saving Project) is an AM process that enables the precise deposition and buildup of clay paste [23]. In this process, the extruded viscous material fully solidifies after the printing process has been executed, e.g., clay objects dry and then are fired in a kiln to reach their optimal structural rigidity.

The experiments were performed on a Delta WASP 40100 clay 3D printer (WASP, Massa Lombarda, Italy) with an LDM extruder. This delta-type printer has 3 DoF enabled through a parallel kinematic mechanism consisting of three arms carrying the actuator attached to three vertical linear carriages. The position of all three arms must be changed simultaneously to obtain a different position in a Cartesian coordinate system. The print volume is cylindrical with diameter and height of 40 cm and 100 cm, respectively. The LDM extrusion system consists of a pressurized 5 L tank that holds the clay and feeds it into the extruder, where it is transported to the nozzle using a spiral screw. The screw is directly attached to the extruder motor (Nema 23 3Nm stepper motor, WASP, Massa Lombarda, Italy) that controls its rotation. Thus, the volume of extruded material depends on the analog parameters of material rheology and tank air pressure and on the digital parameter of screw rotation. For the experiments, a commercially available nozzle with a circular die (inner diameter of 6 mm) was modified by extending the die through a 10 cm long metal tube (inner diameter of 5 mm), which allowed for collision-free material deposition (Figure 2).

*2.3. Parametric Modeling and G-Code*

A design-to-fabrication approach was used to generate the experiments, where the modeling and compiling of the G-code was conducted inside the parametric modeling software Grasshopper 3D in Rhino 7. Surface geometries were modeled, sliced into curves and then divided into geometrical target points. Additionally, travel movements and extrusion values were generated to take advantage of the plasticity of the material. In the case of vertical extrusions, a discrete path consisting of SD was modeled and remapped onto a surface. The various digital parameters were fed into a G-code compiler to generate the print instructions [24].

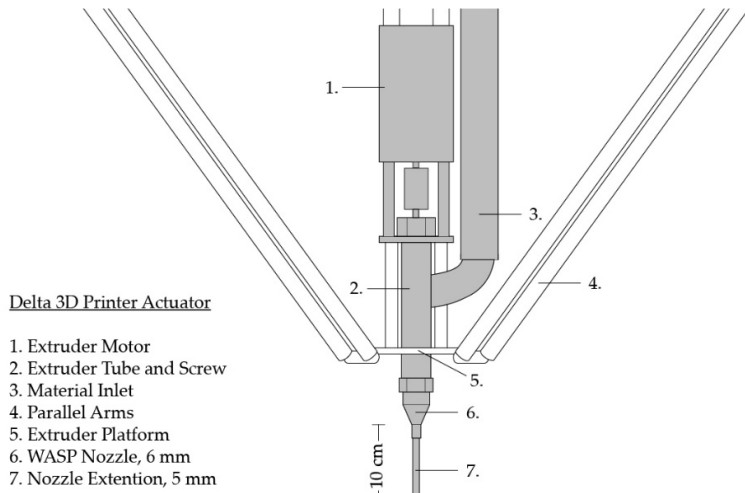

**Figure 2.** Scheme of the used Delta 3D Printer Actuator.

The WASP 3D clay printer is controlled using the G-code programming language, widely used in other automated machine tools, such as CNC milling machines or FDM 3D printers. In conventional FDM and LDM, the G-code is generated with any generic slicer software that slices a digital volume model into horizontal layers, divides it into points and assigns extrusion values that are executed from point to point. The distance between the points defines the resolution of the print, where a good resolution can be achieved when the distance between the points is equal to the nozzle diameter. These instructions are loaded into the 3D printer and executed by the axis and extrusion motors. The extrusion path is the most tangible and easily modifiable aspect of actuator material deposition AM, such as FDM or LDM. However, the control over the extrusion value in the G-code provides digital control over the flow of clay paste by allowing different values for different segments of the model. It enables travel paths where no material is extruded.

A single line of code corresponds to a single action of the machine, whereas a standard line of code for a 3D printer consists of six variables:

- The type of movement that uses the letter address G, e.g., G00 and G01 are the commands for linear movements (without and with extrusion).
- The velocity variable F controls the speed of movement towards the set goal/ coordinate/ point and is set in mm/min.
- The variables X, Y and Z describe the absolute or relative coordinates for the geometrical target point of the movement set in mm.
- The variable E sets the number of rotations of the extruder motor. The WASP 3D clay printer is pre-calibrated by the manufacturer, and an extrusion value corresponding to the distance between two consecutive points (path length) will produce a sufficient volume of material to cover the traveled distance. For example, an extrusion value of 10, corresponding to a rotation of the motor of $45°$, will produce material that can horizontally cover 10 mm. By multiplying the path length using an extrusion factor, the volume of extruded material can be digitally increased or decreased, without modifying the pressure in the clay tank, leading to an object with dynamic extrusion.

For example, to produce a horizontal line of 100 mm length from 0, 0, 0 in X direction at 1000 mm/min, the two G-code lines will look like this:

G01 F1000 X0 Y0 Z0 E0

G01 F1000 X100 Y0 Z0 E100

### 2.4. Spot Deposition

Specialized AM strategies can be created by utilizing the syntax and vocabulary provided by the G-code programming language. In this paper, the basis of the two presented AM strategies is SD. It can be used to create geometries layer by layer, but it also allows for layer-free deposition such as vertical or random deposition. SD is modeled by creating geometrical target points and a separate extrusion value for each target. This is achieved in G-code by separating movement and extrusion in two lines of code, meaning that the 3D printer will sequentially execute them. For example, for two SDs at a 5 mm distance in X direction from each other:

G01 F1000 X0 Y0 Z0 E0

G00 E5

G01 F1000 X5 Y0 Z0 E0

G00 E5

The geometries described in Section 3 were modeled by first creating a single curved shell and then creating layered or semi-random points on the surface. The deposition order of the points in the layered geometry was alternated in each layer, so that there were no travel paths between layers (Figure 3).

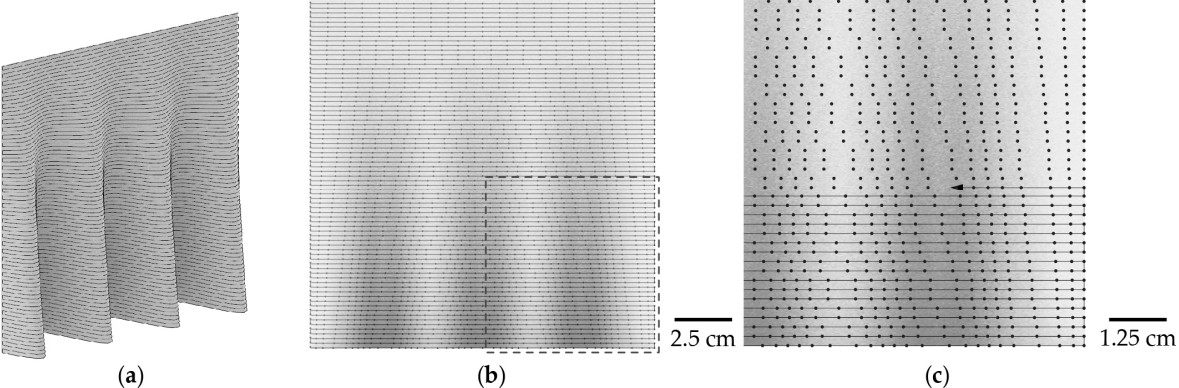

**Figure 3.** Layered SD: (**a**) Isometric model; (**b**) Frontal elevation; (**c**) Geometric goal points and alternating tool-path direction.

A random distribution of points on the digital surface was created for the semi-random deposition strategy. The points were then sorted by finding the ten closest neighboring points and choosing the one with the lowest Z-value. In this way, no SD would be executed in the air (Figure 4c).

Vertical SDs (vSDs) can be created by layering SDs on top of each other (Figure 5a). Vertical SDs were tested with assigned constant and decreasing extrusion values ($E_{base} = E_{top}$ and $E_{base} > E_{top}$). Unlike thermoplastic, where the material hardens shortly after deposition, a continuous vertical extrusion in LDM produces unstable whirling strands (Figure 5c) [25]. SDs assigned descending arithmetic extrusion values produced stable tapered vSDs with conical shape due to the relatively large amount of material deposited at the base (Figure 5a,d). Through an additional horizontal travel path (Figure 5b,d) at the end of the extrusion, targeted deflections could be used to create patterns. The tapered vSDs were generated on a horizontal surface and used to create porous panels (15 × 15 cm). For these experiments, the height of the vertical strands was limited to 30 mm as the standard nozzle was used, which, due to its conical shape and short die, collided with higher vSDs (Figure 2).

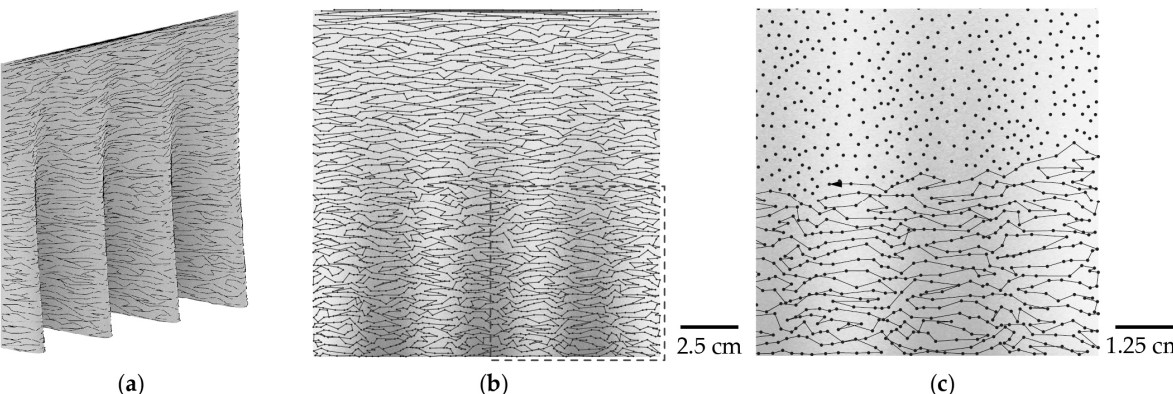

**Figure 4.** Semi-random SD: (**a**) Isometric model; (**b**) Frontal elevation; (**c**) Geometric goal points and tool-path direction.

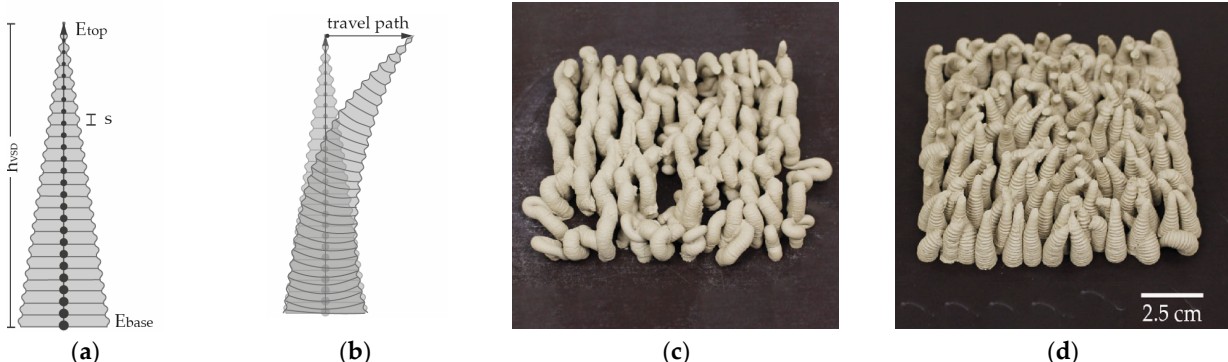

**Figure 5.** (**a**) Vertical SD scheme; (**b**) Deflection scheme; (**c**) Vertical SDs with constant extrusion; (**d**) Tapered vSDs with additional deformation though travel paths.

Higher vSD was enabled through the custom nozzle extension. To find what extrusion (E) value range would produce a geometrically stable vertical strand during and after material buildup, that is also pliable enough to be deformed, a test was conducted considering total height ($h_{VSD}$) and segment height (s). Here, a speed of movement of 33.33 mm/s and segment heights of 0.5 mm, 1.0 mm and 2.0 mm were tested. The experiments showed that an extrusion factor ($f_{base}$) of approximately 33% of $h_{VSD}$ that arithmetically decreases to 1 ($f_{top}$) would produce a stable vertical extrusion (Figure 6).

$$E_{base}, \ldots E_{top} = s \cdot f_{top}, \ldots s \cdot f_{base} \qquad (1)$$

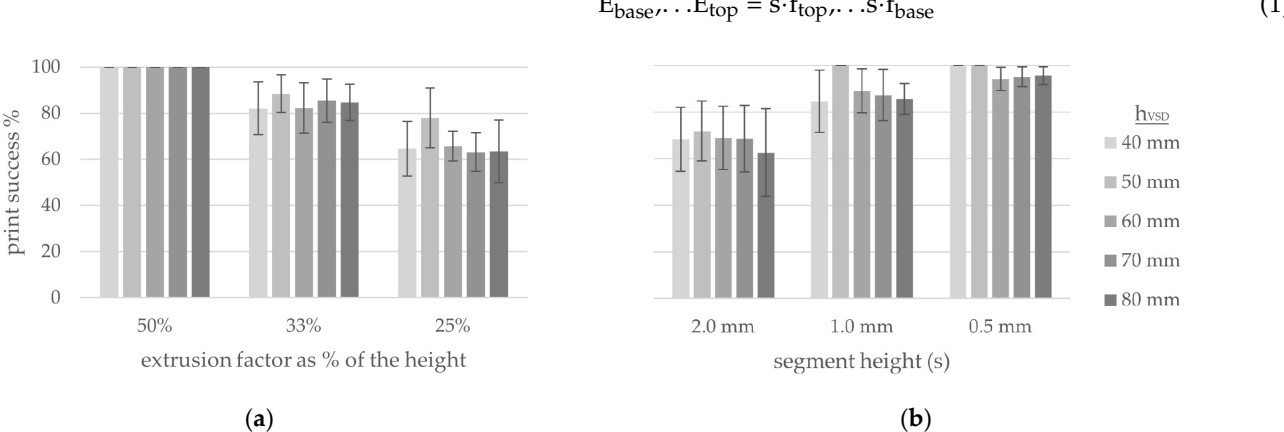

**Figure 6.** Print success of vSD of different heights with decreasing extrusion: (**a**) Extrusion factor ($f_{base}$) as % of $h_{VSD}$; (**b**) Segment height (s).

The vSDs were also tested against continuous vertical extrusion, which exhibited irregular and fluctuating volumes of extruded material due to the uneven translation of the G-code through the 3D printer caused by the rapid changes in extrusion value.

### 2.5. Evaluation of Vertical Face Porosity

Considering additive manufacturing using pastes, the scale of production plays an important role on the main forces that have an impact on the final geometry. On the scale of 0.1–100 mm, where clay pastes are deposited using Direct Ink Writing (DIW), the rheological properties of the used clay paste have a larger influence on the geometry [26,27]. In LDM, where the production scale ranges from 100 to 1000 mm, the self-weight of the extruded strand and the accumulated material weight have a larger influence on geometry, leading to sagging of the material strand during extrusion and height discrepancies due to settling of the geometry [28]. The stability of the geometry is dependent on the tensile and elastic strengths of the extruded clay coil and its compression strength after deposition. Thus, vertical discrepancies of the extruded material are the main challenge for the creation of voids in the vertical dimension. Based on this, the surface porosity and lattice structures produced in this study focused on evaluating the pore area of the main vertical surface compared to its total surface area. This evaluation does not provide a measurement of the total volume of porosity of the produced object; however, it provides comparison with the studies mentioned in Section 1.

## 3. Horizontal Spot Deposition and Extrusion Control

As a basis for SD, a test was conducted to find the right range for extrusion factor parameters, which could enable the stable object buildup. Using the layered modeling method described in Section 2.4, different distances between deposition points (d) and layer heights ($h_L$) were tested (Video S1). The extrusion values were equal to the distances between points. Sample surfaces where the horizontal distance was smaller or equal to the die diameter did not produce a true SD but rather a continuously layered object with a fuzzy surface (Figure 7b). A stable buildup could be produced when the horizontal distance was greater than the die diameter, and the die was cleanly separated from the deposited material through shear force applied by the edge of the die while it traveled to the next goal for SD (Figure 7c,d). Moreover, in these cases, due to the cleaner separation, less of the horizontal movement of the extruder was transferred to the already built-up geometry, leading to less overall displacement during buildup (Appendix A—Table A1). In all samples, the tool-path direction led to visible deformation of the SD.

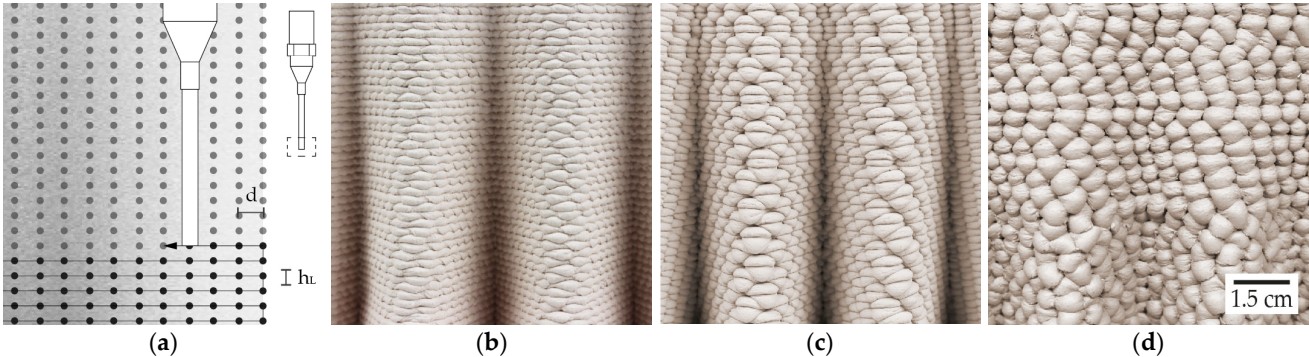

**Figure 7.** Horizontal SD: (**a**) layered SD scheme; (**b**) $h_L$ = 2 mm, d = 4 mm; (**c**) $h_L$ = 2 mm, d = 7.5 mm; (**d**) $h_L$ = 4 mm, d = 7.5 mm.

As the SD strategy does not require the order of the points to follow a path from the smallest to the largest value of the Z-coordinates, a tool path that deposits material at the closest neighboring point with the lowest Z-value could be utilized. The resulting objects exhibited irregular, bumpy surfaces. Also interesting was that, due to material saturation

of the air gaps between each SD, the top third of the surface became thicker compared to the bottom (Figure 8).

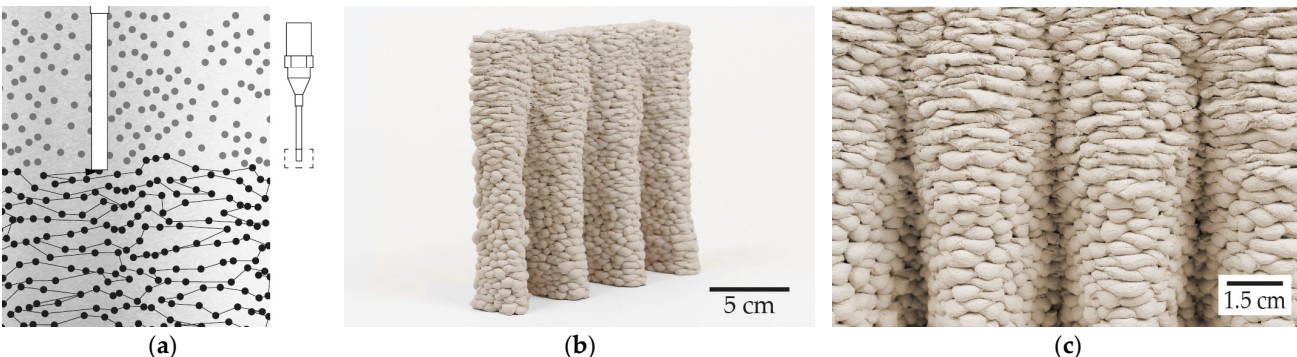

(**a**)      (**b**)      (**c**)

**Figure 8.** Semi-random SD: (**a**) semi-random SD scheme; (**b**) Curved surface produced through semi-random SDs; (**c**) Surface close-up.

By assigning extrusion factors for each point, porous surfaces could be created in layered and semi-random SD and continuously layered material deposition (Video S2). The extrusion factors were generated through sampling of a black-and-white image. In the case of the porous surfaces, the extrusion factors of the points in the black regions were set to 1 and in the white regions were set to 0. It can be noted that the continuously layered porous surface was closest to the original control image with a maximal opening of 8 by 14 mm (Figure 9d). However, in all cases, the holes in the surface were smaller than the modeled white regions. This is a result of the nozzle geometry, as its stopping point is in its center, leading to material being extruded over a half of diameter over the edge. Further discrepancies are caused by the movement of the nozzle, which displaces deposited material in regions where no material should be deposited.

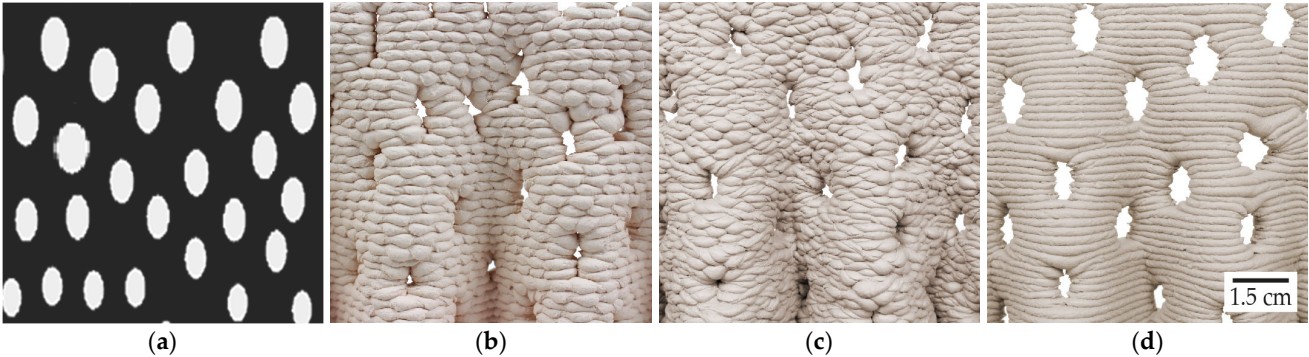

(**a**)      (**b**)      (**c**)      (**d**)

**Figure 9.** Porous surfaces created through the modification of extrusion factors: (**a**) source image; (**b**) horizontal SD; (**c**) semi-random SD; (**d**) continuous deposition.

The plasticity of clay paste enables the over-extrusion of material, where the already deposited material expands to allow for a higher volume of material to be deposited. The sample image (Figure 9a) was also used to create the surfaces with thickened regions, where the extrusion factor of the black was set to 1 and white to a value of 2 to 4 (Figure 10). In the cases where the extrusion value for the white regions is larger than 2, the organic behavior of the clay paste can be observed. Here, the regions with over-extrusion produced thickened regions where the first layers of over-extrusion were incorporated into the previously deposited material and then overflowed outside of the surface (Figure 10b). Using SD, the over-extruded material flowed in the form of a drop instead of being integrated within the already deposited material due to displacement in the normal direction to the surface of the geometry in the previously deposited material caused by the static SD (Figure 10a). In the case where the ratio of fb:fw is 1:2, the effect of over-extrusion is less visible and there is

no drop formation or overflowing pattern (Figure 10c). It can be observed that, similarly to the porous surfaces here, the effects of the over-extrusion do not match the source image. Here, the additional deposited material spreads outside the white contours to create larger, thickened regions.

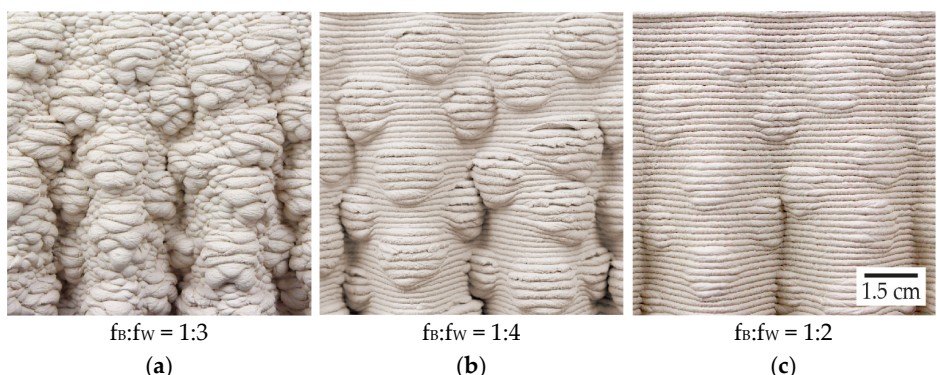

| $f_B$:$f_W$ = 1:3 | $f_B$:$f_W$ = 1:4 | $f_B$:$f_W$ = 1:2 |
|:---:|:---:|:---:|
| (**a**) | (**b**) | (**c**) |

**Figure 10.** Surfaces with over-extrusion: (**a**) semi-random SD; (**b**) continuous deposition; (**c**) continuous deposition.

Considering these results in combination with the results from the layered and semi-random SD described at the beginning of this section, vertical strands of 40 mm were modeled onto the single-curved, digital surface with an additional travel path, which deflects the vSDs away from the face of the surface. Under consideration that, in LDM, the direction of the tool path has a non-trivial effect on the geometry, the bending direction was alternated in each vertical strand [29]. Using this material deposition strategy, the vSDs were successfully stacked on top of each other and produced surfaces with strong reliefs (Figure 11).

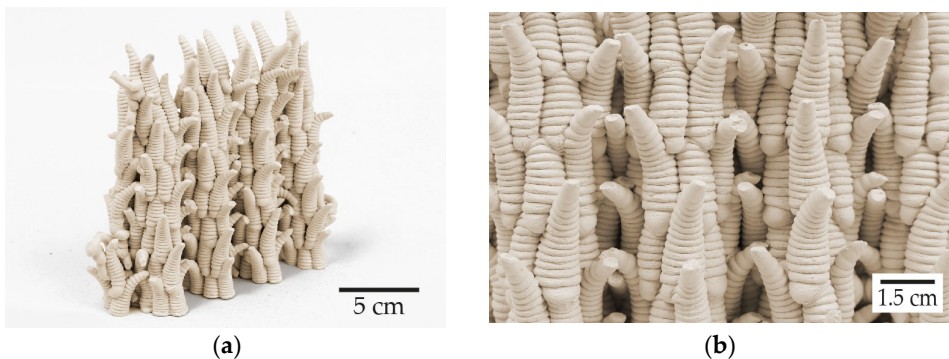

| (**a**) | (**b**) |
|:---:|:---:|

**Figure 11.** Vertical SD and deflection: (**a**) Curved surface produced through vSDs; (**b**) Surface close-up.

## 4. 3D Lattice Structures out of Bent Vertical Extrusions

As observed in Section 2.3, the targeted deflecting of the vertical strands leads to linear instead of point supports that reduce the axial instability and whirling of the following vSDs. Stacking and deflecting the vertical strands in alternating directions achieved a stable buildup process. Considering this, an arc-like geometry with semi-constant width could be produced by bending two tapered vSDs towards each other. This arc system is then utilized to create simple 3D lattice structures with larger openings compared to the porous surfaces produced through extrusion control.

### 4.1. Bending of the Vertical Spot Depositions (vSDs)

A downward bending travel movement was tested in angles ($\alpha$) of 30°, 45° and 60° to evaluate the deformation and the possibility of arc building of vSDs. All vSDs were produced with a tapering extrusion factor of 33% of the $h_{VSD}$ to 1. An angle of 30°

produced a bend of 180° in the geometry, which led to the strand being pressed against the nozzle and sticking to it. A bending angle equal to or above 45° produced a cleaner bend, whereas an angle of 60° produced a base bend ($h_b$) with the height of 16.0 mm (Figure 12). Additionally, a segment height of 2.0 mm produced a higher base bend compared to a vSD with segment height of 1.0 mm. The $h_b$ is measured by extruding vSDs on a horizontal plate and physically measuring the height at the base bend after the execution of the bending travel path and separation from the nozzle. The $h_b$ is of importance for further determining a stable layer height for the stacking of the bent vSDs and ultimately for a stable object buildup.

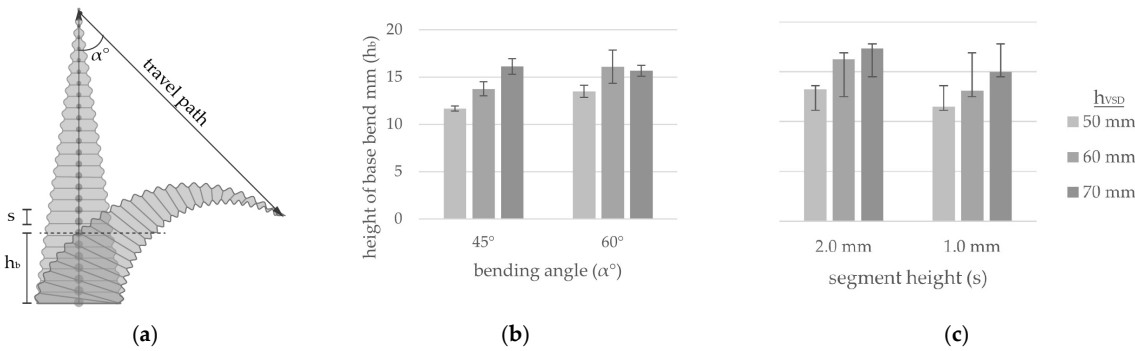

**Figure 12.** Maximum bent height at the base of vSD for: (**a**) Bending scheme; (**b**) Bending angle; (**c**) Segment height.

### 4.2. Interlocking Patterns

The vSDs must be bent parallel to each other to prevent collision. This requirement produces two possibilities for overlapping. The first is the horizontal overlapping of two vertical strands from one layer, and the second is the vertical overlapping of vertical strands from two layers. Using horizontal overlapping, the vertical strands must be rotated around their vertical axis for collision-free deformation. There are different sub-variations for the horizontal folding of the vertical strands, one where one strand of the pair is always facing in the same direction and one where the orientation of the strands is alternated between pairs. In both cases, the direction of deformation is altered in each layer so that the system can stay balanced (Figure 13).

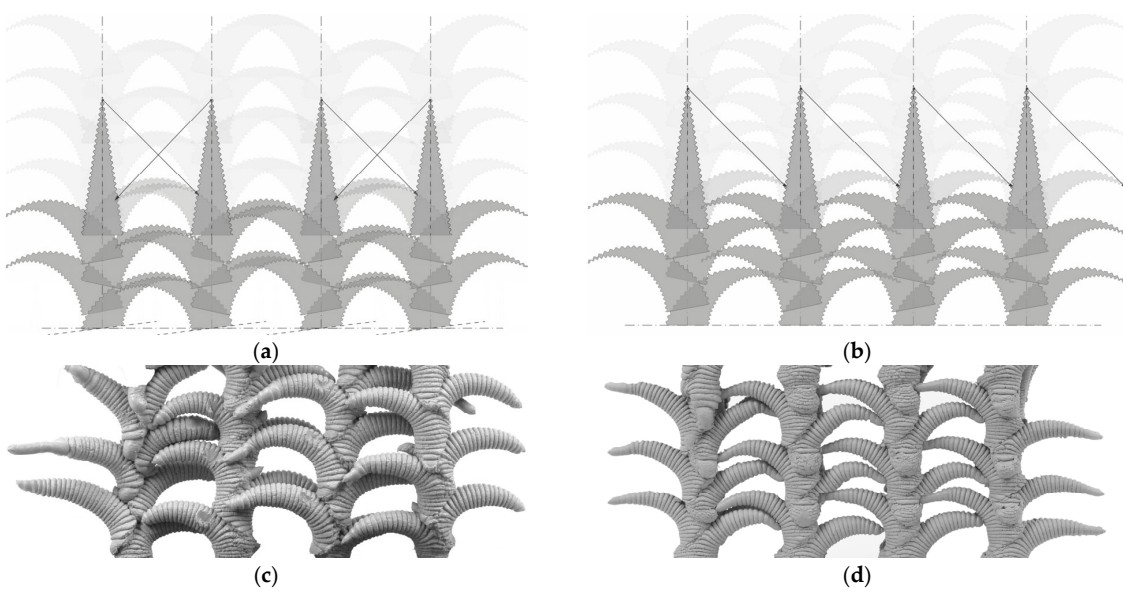

**Figure 13.** Interlocking patterns: (**a**) Horizontal interlocking; (**b**) Vertical interlocking; (**c**) Collage horizontal interlocking; (**d**) Collage vertical interlocking.

### 4.3. Closed One-Shelled Lattice Structures

The viability of the proposed patterns was then tested on a closed surface in the form of a cylinder with a diameter of 15 cm (Figure 14). Closed surfaces have no open edges where fringing of the pattern can occur. The initial interlocking tests used a vSD with a height of 40 mm (see parameters in Table 2). In this case study, the horizontal interlocking pattern provided a stable print, as the nozzle can be cleanly separated from the vertical strand due to the alternating direction of folding in each layer. This ensures that no material accumulation on the nozzle would cause cascading discrepancies leading to a print job failure. Furthermore, the horizontally overlapped strands proved to stick together due to the small ridges created by the SD. The vertical interlocking pattern produced a stable print, but no openings. This case study proved the viability of the proposed interlocking patterns for creating porous geometries. However, the scale of the achieved openings was in the size range of the openings created through image-sampled extrusion control.

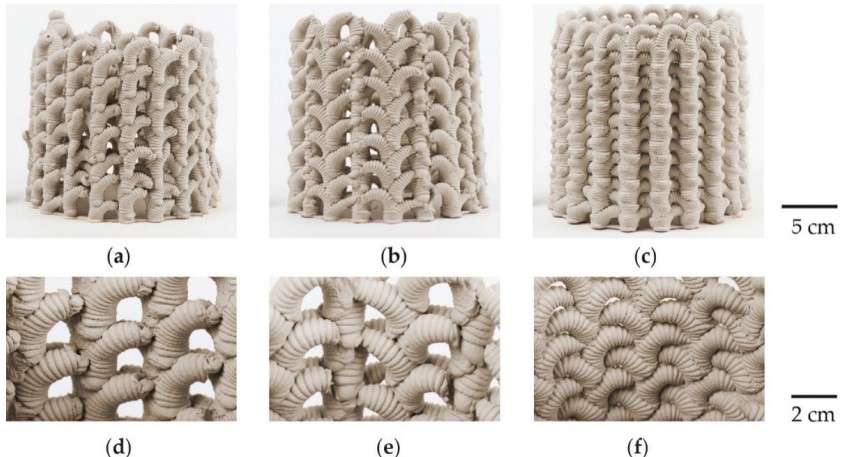

**Figure 14.** Closed one-shelled lattice structures: (**a**,**d**) Horizontal interlocking with same direction of rotation; (**b**,**e**) Horizontal interlocking with alternating direction of rotation of the strand pairs; (**c**,**f**) Vertical interlocking.

**Table 2.** Cylinder parameters of Figure 14.

| Cylinder | $h_L$ * | $D_{VSD}$ * | $\alpha°$ * | $l_{AP}$ * | Print Time |
|----------|---------|-------------|-------------|------------|------------|
| a | 20 mm | 15 mm | 35° | 0.0 mm | 63 min |
| b | 20 mm | 15 mm | 35° | 0.0 mm | 68 min |
| c | 17.4 mm | 15 mm | 35° | 20.0 mm | 60 min |

* For parameter abbreviation, see Figure 15.

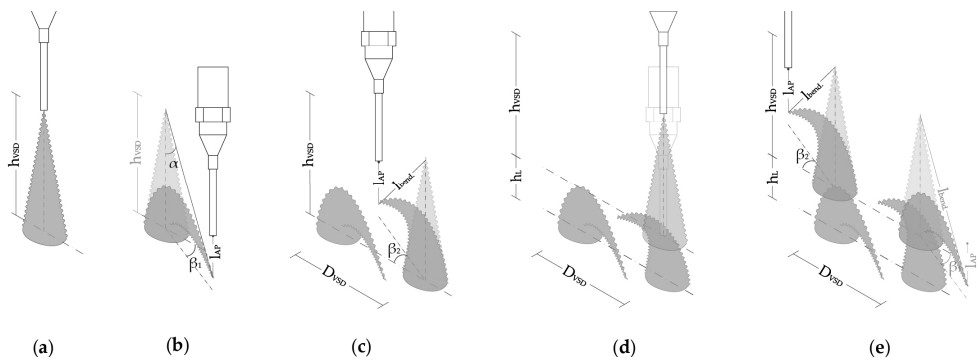

**Figure 15.** Isometric scheme of the parameters for horizontal interlocking: (**a**) vSD; (**b**) bending of vSD; (**c**) second vSD with horizontal interlocking; (**d**) second layer of vSD; (**e**) alternated bending direction of second-layer vSDs.

### 4.4. Opened One-Shelled Lattice Structures

Considering the height and bending test results described in Section 4.2, the potential for porous structures with larger openings was clear. This section presents the results from parameter exploration to determine viable parameters for lattice structures. For this, the parameter space of the horizontal interlocking pattern was explored (Figure 15). The parameters of height of the vSDs ($h_{VSD}$), extrusion factors and movement speed were kept constant. The explored parameters consisted of the layer height ($h_L$), bending angle ($\alpha$), bending path length ($l_{bend}$), the distance between the vSDs ($D_{VSD}$), the two angles for horizontal rotation of the interlocking vSDs ($\beta_1$, $\beta_2$) and an additional vertical travel path used to cleanly separate the nozzle from the bent vSDs ($l_{AP}$) (Table 3).

**Table 3.** Constants and ranges for horizontal interlocking of 26 tested parameter combinations.

|  | $h_{vSD}$ | Segment (s) | Decreasing f * | Speed | $h_L$ | $D_{VSD}$ | $l_{bend}$ | $\alpha°$ | $\beta1$ | $\beta2$ | $l_{AP}$ |
|---|---|---|---|---|---|---|---|---|---|---|---|
| Upper limit |  |  |  |  | 20 mm | 52.5 mm | 80.0 mm | 45° | 19.5° | 14.9° | 60.0 mm |
| Lower limit | 60 mm | 2 mm | 20 to 1 | 2000 mm/sec | 12.5 mm | 30.0 mm | 60.0 mm | 89° | 4.9° | 4.6° | 0.0 mm |
| Mean |  |  |  |  | 17.4 mm | 39.4 mm | 73.7 mm | 57.3° | 11.2° | 8.2° | 19.3 mm |

\* Decreasing extrusion factors.

A bending angle above 60° leads to a higher base bend, but produces only punctual instead of linear interlocking between two strands. Therefore, for a stable buildup, an angle of 45° was used. Stable interlocking between vSDs is crucial for the stability of the deposition and the bending of the consecutive layers. However, material consistency plays an important role in the surface connection behavior of the horizontally interlocked clay strands. The parameter of layer height leads to re-extrusion in the already bent vSDs, which allows for a better bond between the already deposited and to-be-deposited material. Further, the additional material introduced into the bent vSDs applies downward pressure on the base, leading to pre-compression in the already deposited and bent vSDs. In the cases where no material re-extrusion took place, the produced prints collapsed after the first layer. Many of the parameters considered non-viable can be retested using clay paste with another consistency. The discovered viable parameters (see Table 4) produced a geometry with height of around 15 cm, with openings of 30 mm length and 12.5 mm height and with porosity of the object around 19% of the total surface (Figure 16). The height of the produced objects was limited, as the layer buildup was performed in a single vertical plane. This did not provide enough support when forces normal to the vertical plane occurred, which led to a collapse of the total structure.

**Table 4.** Viable parameters for horizontal interlocking.

| $h_L$ | $D_{VSD}$ | $l_{bend}$ | $\alpha°$ | $\beta_1$ | $\beta_2$ | $l_{AP}$ |
|---|---|---|---|---|---|---|
| 12.5 mm | 45 mm | 80 mm | 45° | 8° | 8° | 60 mm |

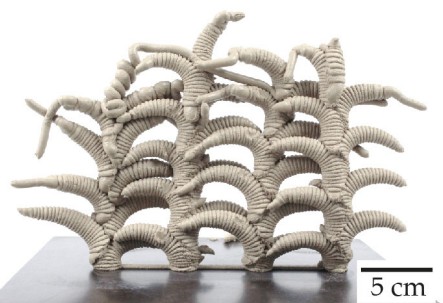

**Figure 16.** Physical geometry of the single-plane horizontal interlocking produced with the parameters from Table 4.

### 4.5. Opened Two-Shelled Linear Lattice Structures

By utilizing the proposed interlocking patterns in three directions, multi-shelled geometries can be created by interlocking vSDs between two shells. The discovered parameters in Section 4.3 were applied to a two-shelled linear geometry where each third layer consisted of horizontally interlocked vSDs connecting the two shells. Both vertical and horizontal interlocking proved viable for the fabrication of two-shelled 3D lattice structures.

The two-shelled geometry with horizontal overlapping had a length of 15 cm, 6 cm width and height of around 24 cm with a printing time of 76 min (Video S3). The face porosity was around 35% with openings of around 25 mm width and 30 mm height (Figure 17). Due to the poor surface connection of the strands along the long side of the geometry, the first column on the back and front separated as one piece from the rest of the geometry during drying.

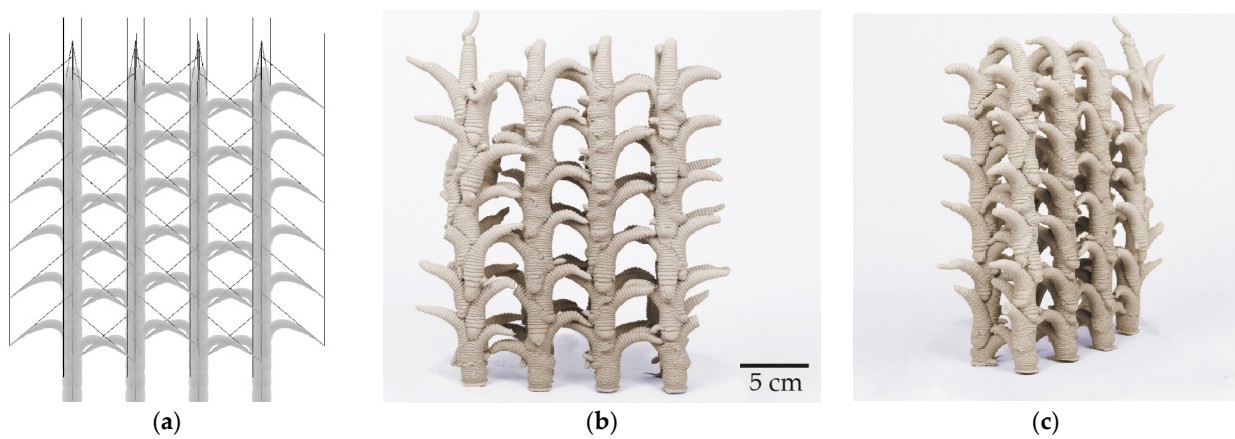

(**a**)      (**b**)      (**c**)

**Figure 17.** Two-shelled linear lattice structure with horizontal overlapping: (**a**) Digital model; (**b**) Front view; (**c**) Diagonal view.

The two-shelled geometry produced through vertical overlapping had a length of 15 cm, width of 6 cm and height of around 20 cm. The face porosity was around 34% with openings of around 25 mm width and 20 mm height (Figure 18). The printing time was 60 min (Video S4). Vertical interlocking proved more suitable for dryer, less sticky clay, where the strand interlocking is not achieved linearly but trough overlap and securing of the tip of the bent vSD in the base of the following layer of vSDs.

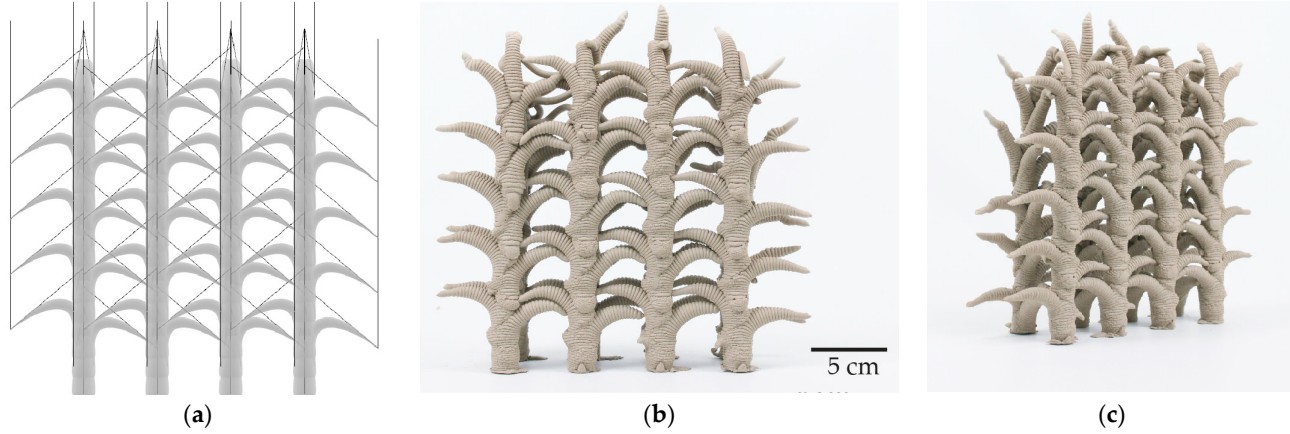

(**a**)      (**b**)      (**c**)

**Figure 18.** Two-shelled linear lattice structure with vertical overlapping: (**a**) Digital model; (**b**) Front view; (**c**) Diagonal view.

### 4.6. Opened Two-Shelled Curved Lattice Structures

Using the horizontal interlocking and the parameters from Section 4.3, an opened two-shelled curved lattice structure was created (Figure 19). The geometry has a length of 30 cm, width of 6 cm and height of 25 cm and was printed for 86 min. However, it exhibits lower porosity than the structures described in Section 4.6 with openings of 10 mm width and 20 mm height and porosity of 20% of the face surface. Due to the curvature of the geometry and the rotation of the bending travel path, there were collisions between the nozzle and the already deposited material. Further, due to the clay shrinkage during drying and the proportion of the structure, it cracked vertically in two along the middle.

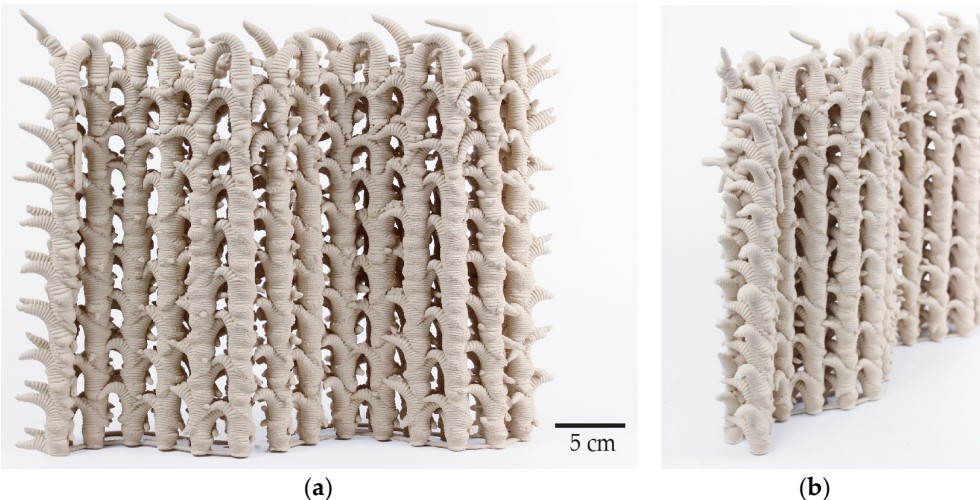

| (a) | (b) |

**Figure 19.** Two-shelled curved lattice structure with horizontal overlapping: (**a**) Front view; (**b**) Diagonal view.

## 5. Discussion

The results of developing two novel additive manufacturing strategies are presented using extrusion control and travel paths. In both cases, the main modeling modes are Spot Deposition (SD) and Extrusion Control (EC), which enable the punctual deposition of a predetermined amount of material. Spot Deposition can be applied horizontally or in a non-layered manner.

The layered, semi-random SD and the deflected vSD allow for the creation of surfaces with strong relief, whereas the semi-random SD and the deflected vSD produce organic non-ordered surfaces without the need for a digital volume model describing the organic surface. In the cases of layered and semi-random SD, the effects can be seen on both sides of the geometry, compared to continuous extrusion effects which are generated on one side of the surface due to their geometrical nature. The execution of the spot material deposition, where the predetermined amount of material is deposited statically, leads to a compression of the material, which does not occur during continuous deposition. This leads to the creation of geometries with less settling under self-weight at a layer height of around half of the nozzle diameter. The downside of using SD to create layered surfaces is the doubling of the job execution time due to the doubling of programmed actions. Using EC, porous single-faced surfaces were created with pores between 3 and 8 mm and over-extruded regions in the 10 to 16 mm range. Black-and-white images as the source of porosity or over-extrusion allow for modeling complex surfaces without geometrical input.

In this study, the deformation of vSDs allowed for the comparatively uncomplicated modeling and fabrication of lattice structures with a maximum height of 25 cm. Material consistency became apparent in generating the lattice structures—material with higher water content, above 22.5wt.%, led to better adhesion in the interlocking pattern, but the layer height was limited to around 25% of the line height, and the distance between two interlocking vSDs was limited to 50% of the line height. Using material with lower

water content of around 21.7wt% allowed for higher vertical extrusion, layer height and distance between columns but hindered the surface connection between two horizontally interlocked strands, which led part of the geometry to separate during drying. In these cases, it can only be stated theoretically that vertical interlocking would be more suitable for clay pastes with lower water content, as in this case, the connection between bent strands depends on the material bond. Thus, horizontal overlapping would be more suitable for clay pastes with higher water content, allowing for better surface connection between two interlocked strands. Cracking occurred through lack of surface connection and geometry having longitudinal proportions. The bending and interlocking of vSDs allowed for the creation of lattice structures with the highest pore size of 20 by 25 mm. A bending angle of 45° allowed for arc-like bending of the strand and cleaner separation between die and nozzle.

It has to be noted that SD was utilized as it proved to produce more exact extrusion compared to continuous material deposition, and a segment height of 2.0 mm allowed for better bending behavior compared to segment heights of 1.0 mm and 0.5 mm. Re-extrusion in a bent vSD led to a pre-compression of the geometry, limited settling during printing and allowed for a layer height consistency with comparatively low height discrepancies between the digital model and the physical object (Figures 17 and 18). Further analysis of the compression strength has not yet been conducted. Finally, it can be stated that the proposed additive fabrication strategy, when compared to previous research, utilizes material behavior as a strength and expands the AM strategies for production of high porosity ceramic elements (Table 5).

**Table 5.** Porosity of the vertical face comparison between reference project and study results.

| Structure | Area of the Vertical Face (cm²) | Achieved Height (mm) | Porosity Normal to the Surface of Vertical Face | Height Deviation from Dig. Model (mm) | Pore Width × Height (mm) |
|---|---|---|---|---|---|
| Spatial Print Trajectory [12] | 330 | 220 | 25% | 60 | 15 × 30 |
| two-shelled lattice with horizontal overlap | 360 | 240 | 35% | 2.5 | 25 × 30 |
| two-shelled lattice with vertical overlap | 300 | 200 | 34% | 10 | 25 × 20 |

Considering these results, further research is possible. Firstly, research on material consistency and its impact on bending and surface connection is important to narrow the field of possible parameters, such as layer height and surface adhesion. Surface adhesion could be improved by modeling an additional travel path that can press two horizontally parallel strands together. Further, a digital exploration of the parameter space by testing for collisions and viable bending parameters based on the material consistency could allow for evaluating designs before production. Using a robot arm with 6 DoF could allow for better control over the strand–die adhesion, separation and collision reduction. By aligning the nozzle die co-axially to the bending path, the deposited material and the material in the nozzle would only be subjected to tension forces. By rotating the nozzle around the axis of the bending path, shear force could be utilized to separate the clay strand from the die. Due to the geometrical similarity of the structures from vertical extrusions to plant growth, growth algorithms could be used to search for further viable structures inspired by organic growth [30].

## 6. Conclusions

Here, a short list of the most important conclusions on the proposed novel AM strategies is presented:

- Three-dimensional lattice structures out of clay paste with face porosity in the range of 25–35% can be fabricated using material plasticity and viscosity in combination with dynamic extrusion and deformations through travel path.
- vSDs and pre-compression of the deposited clay material can allow for the creation of non-massive structures with less height discrepancies during material buildup and drying that enables new structural possibilities.
- vSDs and extrusion control allow the harnessing of the plastic behavior of clay paste for the creation of complex surfaces without volumetric modeling, without much computing capacity and without digital feedback loops.

**Supplementary Materials:** The following supporting information can be downloaded at: https://www.mdpi.com/article/10.3390/ceramics6040124/s1; Video S1: Layered Spot Deposition; Video S2: Semi-random Spot Deposition with dynamic Extrusion Control; Video S3: Two-shelled linear lattice with horizontal overlap; Video S4: Two-shelled linear lattice with vertical overlap.

**Author Contributions:** Conceptualization, V.T. and C.K.; methodology, V.T.; software, V.T.; formal analysis, V.T.; investigation, V.T.; resources, T.H.S.; data curation, V.T.; writing—original draft preparation, V.T.; writing—review and editing, C.K.; visualization, V.T.; supervision, C.K. and T.H.S.; project administration, V.T. and C.K. All authors have read and agreed to the published version of the manuscript.

**Funding:** This research received no external funding.

**Institutional Review Board Statement:** Not applicable.

**Informed Consent Statement:** Not applicable.

**Data Availability Statement:** The data presented in this study are available in this article and the Supplementary Materials/Appendix A.

**Conflicts of Interest:** The authors declare no conflict of interest.

## Appendix A

In this appendix, the explored parameters for horizontal SD are presented. The visual evaluation was performed by comparing the performance of all samples using the given evaluation factors and ranking them at even increments.

**Table A1.** Horizontal SD parameters and visual evaluation.

| | Parameters | | | Visual Evaluation | | | |
|---|---|---|---|---|---|---|---|
| Test No. | Speed (mm/s) | Layer h (mm) | Resolution (mm) | Clean Nozzle Separation | Displacement | Spot Merging | Collapse |
| 1 | 20 | 2 | 2 | false | 0 | 1 | false |
| 2 | 20 | 2 | 4 | false | 1 | 1 | false |
| 3 | 20 | 2 | 7.5 | true | 0.5 | 0.5 | false |
| 4 | 20 | 2 | 10 | true | 0.25 | 0.25 | false |
| 5 | 20 | 4 | 4 | false | 0.75 | 1 | false |
| 6 | 20 | 6 | 2 | false | 0 | 1 | true |
| 7 | 20 | 4 | 7.5 | true | 1 | 0.5 | false |

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
