# Peer review of "Dynamic Extrusion Control in Spot Deposition Modeling for Porous 3D Clay Structures"

_ceramics, doi:10.3390/ceramics6040124_

Round 1

Reviewer 1 Report

This is an interesting paper, and the work is of interest. However, the manuscript is clearly written, but I have some problems with the presentation of the data. This should not be too difficult for the authors to address:

 1. Why did the authors use Spot Deposition Modeling and not Direct ink writing or material extrusion according to ASTM 52900:2021?

2. The study was based on the advantage of the viscous state and tensile strength of the extruded clay; however, the authors did not present any Viscosity or rheology tests on their printed inks?

3. Please could the authors report the standard deviation of the measured values?

Reviewer 2 Report

The results of this study are very interesting. However, for a non-specialist in additive manufacturing of ceramics, this report is difficult to read as many concepts are not sufficiently introduced, and the originality of the achieved results is hard to appreciate. Further, the paper does not focus enough on relationships between process parameters, ceramic porosity and clay paste printability or viscoelastic properties, in a quantitative and scientifically sounded manner. This is however a prerequisite for aligning with the scope of the journal

1.The introduction should be revised to better assess the novelty of both approach and results of this study. Lines 418-422 should be shifted to the introduction, and porosity concepts should be developed in terms of quantitative assessment for each structure referred in ref. [6-7]. Further, given the amount of studies on this topic (see e.g. https://doi.org/10.1016/j.compositesb.2021.109249), the introduction should be augmented to other relevant references on the direct writing of clay for porous ceramics. Maybe specific application should be pick up for a quantitative target of porosity and product's dimensions.

2. What is “dynamics extrusion” (line 40) or “dynamic extrusion” (line 66). Please define and differentiate from direct extrusion of paste or material extrusion, which is the generic term for 3D direct writing of materials.

3. Line 100: material consistency. Please define this property scientifically. Is this about viscosity, elasticity, structural recovery time, yield stress, etc.?

4. Lines 135-139: how does the quantity of extruded material depend on the material rheology? How does the printer take this into account in the calibration?

5. Lines 180-186: this part needs to be revisited.  Equation 1 is not clear (what is s in sftop?); in figure 6, what are the 40,..., 80mm? Are these hsvd? If yes, please add. Same comment for fig 12. Also in fig 6, how do we see stable extruded strand? Is it dimensional or time stability?

6. Line 200: How is die separation controled? That is, what is the printing parameter that controls the die separation from the deposited spot? What is the material property?

7. Line 201: please define vibration, e.g. by pointing arrows in the figures?

8. Line 209: please explain what is the reason for material saturation.

9. Figure 8 is not referenced in the text.

10. Lines 219-220: please explain why. Is it due to material properties? Printing parameters?

11. Line 226, “was set to 1”. Is this wrong? In fig 10b, fb is 0.8. Also, the ratio between fb and fw seems to be a key parameter: please discuss figs 10b and 10c accordingly.

12. Line 232, “oscillations”. Please define "oscillation". Is this oscillation in time, along a direction, and oscillation of what geometrical parameter on the construct?

13. Line 239: section 2.5 does not exist. Please correct.

14. Line 256. Please add "alpha" after bending. How are these bending angles related with the travel path between each layer in fig5?

15. Line 261. It is not clear what is meant by "peak". Also the base bend height needs to be clearly defined in the text. How is it measured? Note that the scheme in 12a is not clear on defining hb, and how it is measured.

16. Line 319 and through the whole study: how is porosity of printed object assessed? How does this measurement compare with porosity measurement in other studies?

17. Line 320. Please rephrase, as both instability and linear geometry are not clear at this point.

18. Line 399. Re-extrusion and pre-compression concepts should be developed. As it stands in the text, this is not clear at all.

19. Lines 413-414: please rephrase.

Table A1: what is resolution (in mm)? This printing parameter is not defined in the text. Same for build-up stability. Visual evaluation: please explain how numbers are defined.

see above for rephrasing few lines.

Round 2

Reviewer 1 Report

The authors have successfully addressed all comments.

Author Response

Dear Reviewer 1,

we are glad that we have fully addressed your revisions and thank you for the approval for publishing.

Best regards,

Vesela Tabakova

Reviewer 2 Report

Authors positively addressed all issues raised during the review. As a result, this paper has improved. A last comment now deals with the newly added section 2.5: “In LDM…..the role of gravity has a stronger influence on geometry [28]”. If gravity is at play and gives way to sagging, then the rheology of the printed layers is the main issue. This needs to be contrasted with the rheology of the printing (extrusion). But still, rheology of the clay material is central and the sentence reproduced above needs to be changed accordingly.

Author Response

Dear Reviewer, 2,

Thank you for your revisions. The last comment was addressed by reformulating the paragraph, by underscoring the importance of material weight in the system, and more clearly defining its effects and the corresponding material properties that counteract them. Thus, the term ‘sagging’ was used in the context of deformation of the extruded coil and the term ‘settling’ was used to describe height discrepancies in the physical geometry due to the accumulation of weight (Sagging – settling 53, Sagging – settling 80, Sagging – settling 433,Sagging – settling 461). Further sagging was connected to the material properties of tensile strength and elasticity and settling was connected to the material properties of compression strength.

I hope this gives enough clarity to on the phenomena that have influence on the geometry in the scale of LDM.

We hope that we have addressed your comment and that you will give your approval to the publication. 

Respectfully,

Vesela Tabakova